# Heavy khat (Catha edulis) chewing and dyslipidemia as modifiable hypertensive risk factors among patients in Southwest, Ethiopia: Unmatched case-control study

**Meron Hadis Gebremedhin**[1]*, **Eyasu Alem Lake**[2], **Lielt Gebreselassie Gebrekirstos**[3]

**1** School of Medicine, College of Health Sciences and Medicine, Wolaita Sodo University, Wolaita Sodo, Ethiopia, **2** Department of Pediatrics and Child Health Nursing, Wolaita Sodo University, Wolaita Sodo, Southern Ethiopia, **3** Department of Maternity and Reproductive Health Nursing, College of Health Science and Medicine, Wolaita Sodo University, Wolaita Sodo, Ethiopia

* merykiyaggg@gmail.com

## Abstract

### Background

The burden of hypertension is more devastating in low-and middle-income countries, including sub-Saharan Africa than in high-income countries. Among the modifiable risk factors, dyslipidemia and khat chewing were expanding at an alarming rate in Ethiopia but were still underestimated. Thus, this study aimed to assess heavy khat (*Catha edulis*) chewing and dyslipidemia as modifiable hypertensive risk factors among patients in the southwest, Ethiopia.

### Methods

A facility-based case-control study was conducted among 136 cases and 270 controls from May 15 to July 30, 2017. A consecutive sampling technique was used in this study. Epi data version 3.1.1 and SPSS version 21 were used for data entry and analysis. Descriptive statistics and bivariate and multivariate logistic regression analyses were performed. Both crude and adjusted odds ratios and 95% confidence intervals were reported.

### Results

The majority of the cases had a total cholesterol to high-density lipoprotein ratio of >5 (72.1%). The odds of hypertension increased among participants who had attended no formal education [AOR = 2.25, 95% CI: (1.05–4.82)], history of alcohol consumption [AOR = 5.93,95% CI:(3.11–11.30)], moderate khat chewing [AOR = 3.68, 95% CI:(1.69,8.01)], heavy khat chewing [AOR = 18.18, 95% CI: (3.56–92.89)], mild intensity physical activity [AOR = 3.01, 95% CI: (1.66–5.47)], type of oil used for food preparation [AOR = 2.81, 95% CI: (1.49–5.28)], and dyslipidemia [AOR = 6.68, 95% CI: (2.93–15.23)].

**Data Availability Statement:** All relevant data are within the paper and its Supporting Information files.

**Funding:** No funding received but Jimma University paid per diam for data collectors.

**Competing interests:** No competing interests.

**Abbreviations:** AOR, Adjusted odds ratio; BMI, Body mass index; COR, Crudes odds ratio; CVD, Cardiovascular disorder; HDL, High density lipoprotein; LDL, Low density lipoprotein; TC, Total cholesterol; TG, Triglycerides; WHO, World health organization.

## Conclusion

The study showed that modifiable risk factors were the major factors associated with the development of hypertension. The findings of this study highlighted that health education is needed to focus on physical exercise, quitting excess alcohol consumption, quitting khat chewing by giving special emphasis to those who had no formal education. In addition, consideration should be given to a healthy diet free of cholesterol and unhealthy behavior.

## Introduction

Elevated blood pressure (BP) is the leading cause of mortality and morbidity related to cardiovascular disorders worldwide [1]. Globally, it is estimated that one billion adults live with hypertension; this figure is predicted to be more than a 1.5billion by the year 2025 [2]. Currently, the burden of hypertension has become more devastating in low- and middle-income countries, including sub-Saharan Africa (SSA), than in high-income countries [3]. The prevalence of hypertension in SSA is estimated to be approximately 30% [4]. As treatment and control of hypertension are low in this area, the high burden of hypertension in these nations is likely to have an increased risk of more morbidity and mortality from potentially preventable complications such as stroke, myocardial infarction, and renal failure [5]. Worldwide, over 50% of the 17.4 million annual deaths were caused by cardiovascular diseases that attributed to hypertension. Moreover, at least 45% of deaths due to heart disease and 51% of deaths due to stroke were related to hypertension [2]. Africa, including the rural population, needs more consideration of cardiovascular health by responsible authorities [6].

Hypertension is one of the most common public health burdens in Ethiopia. A meta-analysis conducted in Ethiopia reported the prevalence of hypertension to be 19.4% [7]. These may be related to the changing lifestyle of the Ethiopian population due to urbanization and demographic transition [8]. Empirical findings in this country revealed that various risk factors were associated with hypertension. These risk factors were age, obesity, family history of hypertension, smoking, diabetes, alcohol consumption, inadequate intake of fruit and vegetable, excess salt use, and not continuous walking for at least 10 minutes per day [9–12].

Among the modifiable risk factors of hypertension, khat chewing is the one. Khat (Qat, Kat and Miraa, *Catha edulis*) is a dicotyledonous evergreen flowering tree. It is a member of Celastraceae that grows in equatorial climates, mainly in the horn of Africa and the Arabian Peninsula [13]. Approximately, 20 million people worldwide are believed to be khat chewers, which were previously localized in East Africa and the Arabian peninsula [14]. Chewing is the fundamental mechanism by which the fresh leaves of khat were chewed slowly for several hours [6]. A study in Yemen reported a rise in blood pressure with the duration of khat chewing [7] but another study failed to associate [15]. When we compare khat chewers to non-chewers, the prevalence of hypertension and diastolic blood pressure is high among chewers [10].

Ethiopia is one of the five countries that cultivate khat. In Ethiopia, though efforts have been made to identify the risk factors for hypertension and to overcome its effect, the prevalence of HTN and its risk factors have not decreased [16]. Despite some studies uncovering chewing khat repeatedly and frequently results in increased blood pressure and development of myocardial infarction [17–20], most of the previous studies in Ethiopia neglected to investigate the association between khat chewing, dyslipidemia, and hypertension.

Since there is growing evidence on the prevalence of hypertension and khat chewing [20], there is a need to investigate their association for better intervention. Most previous studies

did not investigate the amount of khat chewed, dyslipidemia, and hypertension in a case-control association. Moreover, it was hard to reach more studies that included dyslipidemia and khat chewing as risk factors for hypertension. Thus, this study aimed to assess modifiable risk factors for hypertension among patients in Southwest Ethiopia, including the most dominant factors (khat chewing and dyslipidemia), applying unmatched case-control study. A clear understanding of such factors is crucial for building responsive interventions by the responsible bodies.

## Materials and methods

### Study setting, study period and study design

A facility-based unmatched case-control study was conducted from May 15th to July 30th, 2017, at Jimma University medical center (JUMC), found in Jimma, southwest Ethiopia. Jimma zone is located 600 km southwest of Addis Ababa, Ethiopia's capital. As the information gained from the JUMC website (http://www.ju.edu.et), JUMC is the only referral hospital in the Zone, providing services for approximately 15,000 inpatients and 160,000 outpatients per year with a catchment population of about 15 million people.

### Ascertainment of cases and controls

The source population comprised all adults (aged 18 years and above) who attended outpatient departments in JUMC. Cases were patients already diagnosed with hypertension by a physician or those taking anti-hypertensive drugs during the study period. Controls were patients attending JUMC with no history of hypertension and whose blood pressure was <140 mmHg/ <90 mmHg during the study period.

**Inclusion criteria for cases.** patients who are known hypertensive patients diagnosed by a physician (diagnosed with (BP $\geq$ 140/90)) and on anti-hypertensive treatment and attended services during the study period which were selected consecutively from the follow-up clinic where all patients with chronic illness attend this clinic. Two hospital-based controls were selected from the same medical ward that sought other services and proved to be free from hypertension (diagnosed as normal BP and confirmed by three re-measurement of BP).

**Exclusion criteria for cases and controls.** Pregnant women, a client with renal disease, mentally unstable or critically ill, and unable to respond were excluded from both groups.

### Measurement variables

**Outcome variable.** The outcome variable for the study was hypertension. Patients were categorized as cases if they were diagnosed with hypertension by a physician or those taking anti-hypertensive drugs during the study period.

**Arterial blood pressure.** blood pressure was measured three times according to a standardized protocol using an automatic oscillometric method (dynamap). It was obtained from the left arm in a seated position using a standard mercury sphygmomanometer BP cuff. Participants were asked if they had hot drinks, smoke cigarettes, stressed or had vigorous-intensity physical activity before measurement and then waited for 30min. Finally, the average of the readings was considered as the final BP of each participant. The same blood pressure recorder was used for the overall study population. Hypertension was defined as systolic BP>140 mmHg and/or diastolic BP>90 mmHg or reported use of regular anti-hypertensive medication(s). As the 6th Joint National Committee categorized hypertension as stage one and stage two, it also classified those patients taking the anti-hypertensive drug in stage two [21].

**Independent variables.** The independent variables were conceptualized based on the WHO standard questionnaire [18] and previous similar studies [8–11, 19] and then clustered into four sets of factor characteristics. The independent variables were socio-demographic factors such as educational, occupational, religion, income, and oral contraceptive use history. Non-modifiable factors such as age, sex, family history of hypertension and modifiable lifestyle determinants such as smoking status, history of alcohol consumption, khat chewing, intensity of physical activity, dyslipidemia and anthropometric measurements such as BMI. Heavy khat chewers where those who had a history of khat chewing of two or more bundles of khat in one chewing session, moderate chewers were those who chew almost a bundle of khat and those mild chewers were those who chew less than one bundle of khat for one chewing session at least for more than six months. Since Ethiopian drinks such as "Tella, Areki, Teji, and borde "are the main types of alcoholic drinks in the study area; alcohol consumption was defined as the use of these alcohols in addition to the standard ones. In this study packed (palm) oil was packed in the jar "Chef, Hayat, Viking". Physical activity is considered to be at least 150–300 minutes of moderate-intensity aerobic physical activity; or at least 75–150 minutes of vigorous-intensity aerobic physical activity or an equivalent combination of moderate and vigorous-intensity activity throughout the week [22, 23]. Lipid profiles (Total cholesterol, HDL, LDL, and triglyceride) were determined and dyslipidemia was classified according to the American heart association, and the TC/HDL ratio was >5 [18].

## Sample size determination and sampling technique

The sample size was calculated using Epi info version 3.1 with the following assumptions; 20.9% proportion of exposure (history of cigarette smoking) among control groups with an odds ratio of 2.06 [12], 95% confidence interval, 80% power, and 2:1 control to case ratio. After adding a non-response rate of 10% total sample size was 406 (136 cases and 270 controls). A consecutive sampling technique was used to recruit the required sample sizes for the cases. The control groups were also selected consecutively after case selection had been completed until the calculated sample size was achieved. Selected study participants were interviewed upon their exit from the chronic illness clinic and to avoid overlapping we put a mark on the patient's card.

## Data collection procedure and quality management

Data were collected using pre-tested interviewer-administered structured questionnaires, adapted with modifications from the WHO standard questionnaire [24] and previous studies [11, 12, 17] according to the study objective and local context. Doctors in internship facilitated data collection, six BSc nurses, and two laboratory technicians with extensive experience in data collection collected the data and analyzed it, respectively. The final questionnaire has four sub-parts; socio-demographic characteristics, behavioral characteristics, biochemical characteristics, and nutritional characteristics. Item questions were checked for reliability and internal consistency using Cronbach's alpha coefficients. The tool was translated into the local language (Amharic & Afaan Oromoo) and subsequently translated back to English by different language experts to check the consistency and quality of the translation. (S1 Questionnaire) Study participants were interviewed on their exit from the chronic illness clinic in a private setting after a deep discussion that removed their doubts and cleared their confusion. Before administration of the questionnaire, it was reviewed by two senior experts and pre-tested on 5% of the sample size in the nearby Shenen Gibe hospital. All required revisions were made to the study tool based on the pre-test and expert comments. Before the actual data collection day, two-day intensive training on the aim of the study and sampling procedures was provided

to the enumerators. The supervisors (PI and co-author) underwent routine checkups for completeness and scientific soundness. To minimize measurement error anthropometric measurements were taken after prerequisites to avoid error that is, weight was measured after the scale pointer was checked at zero, and subjects wore light clothes and stood straight and unassisted in the center of a balance platform. Height was also measured after participants were requested to remove their shoes, stood erect, a position at the plane with feet together and knee straight. The heels, buttocks, and shoulder blades were made straight against the stadiometer's vertical stand. Instrument calibration and random auditing were performed; measurements were taken twice, and finally, height to the nearest 0.1cm and weight to the nearest 0.1kg was taken. Anthropometric measurements were translated according to the WHO Steps guidelines. Participants with a BMI lower than 18.5 kg/m$^2$ were considered as underweight; between 18.5 and 24.9 kg/m$^2$ as normal; between 25.0 and 29.9 kg/m$^2$ as overweight and 30.0 kg/m$^2$ and above as obese [25]. Medical records and consultation with the person in charge of the patient were the gold standards for identifying cases and controls.

## Sample collection and biochemical analysis

After overnight fasting, participants conducted the interviews and anthropometric measurements and provided 5 ml of venous blood. Then, the sample was centrifuged for 5 minutes at 4000 revolutions per minute and stored at-80˚C for subsequent biochemical tests. Around 2.5 ml of pure serum sample was separated into a Nunc tube. The samples were analyzed using a Mindray BS-200 chemistry analyzer (Shenzen Mindray Bio-Medical Electronics Co. Ltd) according to the manufacturer's instructions. Before sample analysis, the machine was checked using controls and blank daily. The sample was analyzed to determine the lipid profile of the four parameters (triglyceride, high-density lipoprotein, low-density lipoprotein, and total cholesterol); and dichotomized the parameters and found that there was a risk factor for dyslipidemia (total cholesterol to high-density lipoprotein ratio >5 [26].

## Data processing and analysis

The data were entered into Epi-data version 3.1 and exported to SPSS version 21 for analysis. It was explored to check for outliers, missing data, and assumptions. A chi-square test was used to assess each independent variable's association with the outcome variable. Descriptive statistics and cross-tabulation were calculated. Bivariate and multivariate logistic regression analyses were also performed. Significant variables with a p-value of ≤0.25, in bivariate analysis, were retained for further consideration in multivariate logistic regression to control for confounders. This is basically to compensate for the power of the test because negative findings (i.e. p > 0.05) may be due to inadequate power [27, 28]. Finally, multivariable logistic regression was performed to control for possible confounding effects of variables such as (BMI, behavioral factors and dyslipidemia). Odds ratios and 95% confidence intervals were computed, and a p-value of less than 0.05 was used to determine the cut-off points for statistical significance.

## Ethical approval and consent to participate

Jimma University institute of health science and the ethical review committee approved the study. All the study participants were informed about the purpose of the study, their right to refuse, and ensured confidentiality and verbal and written consent was obtained before the interview. No personal details were recorded or produced on any documentation related to the study and privacy was assured. At the end of each interview and measurement

**Table 1. Socio-demographic characteristics of study participants, Southwest Ethiopia, 2019.**

| Variable | | Case No (%) | Control No (%) | Total No (%) | $X^2$ | p-value |
|---|---|---|---|---|---|---|
| **Sex** | Male | 70(51.5%) | 143(53.0%) | 213(52.5%) | 0.08 | 0.78 |
| | Female | 66(48.5%) | 127(47.0%) | 193(47.5%) | | |
| **Age category (years)** | <35 | 28(20.6%) | 61(22.6%) | 89(21.9%) | 0.26 | 0.88 |
| | 35–55 | 50(36.8%) | 94(34.8%) | 144(35.5%) | | |
| | >55 | 58(42.6%) | 115(42.6%) | 173(42.6%) | | |
| **Place of residence** | Urban | 82(60.3%) | 156(57.8%) | 238(58.6%) | 0.24 | 0.63 |
| | Rural | 54(39.7%) | 114(42.2%) | 168(41.4%) | | |
| **Ethnicity** | Oromo | 96(70.6%) | 184(68.1%) | 280(69.0%) | 11.23 | 0.47 |
| | Amhara | 17(12.5%) | 34(12.6%) | 51(12.6%) | | |
| | *Dawuro*, *Kafa*, *Yem* | 23(16.9%) | 52(19.26%) | 75(18.47%) | | |
| **Marital status** | Married | 106(77.9%) | 217(80.4%) | 323(79.6%) | 0.33 | 0.57 |
| | Single/divorced/ widowed | 30(22.1%) | 53(19.6%) | 83(20.4%) | | |
| **Religion** | Orthodox | 31(22.8%) | 77(28.5%) | 108(26.6%) | 3.44 | 0.33 |
| | Muslim | 85(62.5%) | 156(57.8%) | 241(59.4%) | | |
| | Other** | 20(14.7%) | 37(13.7%) | 57(14.03% | | |
| **Educational status** | No formal education | 78(57.4%) | 113(41.9%) | 191(47.0%) | 14.66 | 0.001* |
| | Primary education | 34(25.0%) | 61(22.6%) | 95(23.4%) | | |
| | Secondary and above | 24(17.6%) | 96(35.6%) | 120(29.6%) | | |
| **Occupation** | Farmer | 48(35.3%) | 74(27.4%) | 122(30.0%) | 3.11 | 0.37 |
| | Government employee | 25(18.4%) | 49(18.1%) | 74(18.2%) | | |
| | House wife | 34(25.0%) | 76(28.1%) | 110(27.1%) | | |
| | Other*** | 29(21.3%) | 71(26.3%) | 100(24.6%) | | |
| **Income (1dollar = 40birr)** | Low | 77(56.6%) | 132(48.9%) | 210(51.5%) | 2.80 | 0.25* |
| | Middle | 32(23.5%) | 66(24.4%) | 98(24.1%) | | |
| | High | 27(19.9%) | 72(26.7%) | 99(24.4%) | | |
| **OCP use** | Yes | 43(65.2%) | 62(48.8%) | 105(54.4%) | 4.67 | 0.03* |
| | No | 23(34.8%) | 65(51.2%) | 88(45.6%) | | |
| **Family hx of HPN** | Yes | 30(22.1%) | 53(19.6%) | 83(20.4%) | 0.33 | 0.57 |
| | No | 106(77.9%) | 217(80.4%) | 323(79.6%) | | |

Note: Other

*p≤0.25

** protestant, Catholic, Adventis

***Student, merchant, Self-Employed, hx; history, ocp; oral contraceptive; HPN; hypertension.

procedure, we created awareness of risk factors of hypertension and aggravating factors for the cases.

## Results

### Socio-demographic characteristics of study participants

A total of 406 participants (136 hypertensive cases and 270 non-hypertensive controls) were participated making, a response rate of 100%. Almost half of the cases 70(51.5%), and 143 (53.0%) controls were male. A large proportion (42.6%) of both the cases and controls fell within the age group of above 55 years. The mean ± SD age for the cases and controls were (52.65±13.09) years and (50.95±13.55), respectively. More than half of the cases and almost half of controls did not attend any formal education (*p<0.001*) and had low monthly income *p = 0.25* Table 1.

## Behavioral, biochemical and nutritional characteristics

Ninety-six (70.6%) cases and one hundred eight (43.7%) controls had a history of khat chewing during their lifetime (p<0.001). A large proportion of cases chew one bundle of khat during one chewing session compared to controls (39.0% vs. 13.3%, (p<0.001)) of which (38.5%) of cases and 36.4% chew khat daily. Most of the reasons for chewing were religious prayers, to increase social interaction, addiction, and to stay awake. More cases 56.6% than controls 26.3% drank alcohol (p<0.001). Among alcohol users, 36.4% of cases and 46.5% of controls drank "Tella", "Tejj", "areki" and beer three times a week. More than half (57.3%) of the cases and (63.0%) of the controls drank alcohol for over ten years. Only 38(27.9%) cases and 40 (14.8%) controls had a previous history of smoking cigarettes (p = 0.002), and only 5.3% of cases and 2.5% of controls were smokers during the time of data collection. Moreover, 93 (68.4%) of cases and 102(37.8%) controls had moderate and vigorous physical activity, p<0.001). More than half of the cases used oral contraceptives compared to controls (65.2% vs. 48.8%, *p = 0.03*).

As shown in Table 2, the total cholesterol levels above 200mg/dl among the two groups were significantly different (58.1% vs. 33.7%, p<0.001). Among the cases, nearly three out of four (72.1%) had dyslipidemia (TC/HDL ratio >5) and (23.7%) of controls. In most of the cases, 77.2% and 63.7% of controls used packed (palm) oil for food preparation. Regarding anthropometric measurements, most of the study participants had a normal body mass index range (18.5–24.9, p = 0.12) Table 2.

## Risk factors for hypertension

Binary logistic regression was performed to determine the association between the dependent and independent variables. Variables that were transferred from the bivariate analysis to the multi-variable analysis were; educational status, income, history of smoking, history of alcohol consumption, history of khat chewing, amount of khat (bundle) chewed per session, the intensity of physical activity, type of oil used for food preparation, salt consumption, TC, HDL level, LDL level, TG level, dyslipidemia, and BMI. After adjusting for confounders; educational status (no formal education), history of alcohol consumption, amount of khat chewed per session, the intensity of physical activity, type of oil used for food preparation and dyslipidemia were significantly associated with hypertension.

Having no formal education increases the odds of developing hypertension by 2.25 times (AOR = 2.25(1.05–4.82) as compared to controls with primary or secondary and above educational status. Similarly, those with a history of excess alcohol consumption had five times (AOR = 5.93(3.11–11.30)) higher odds of developing hypertension than those with no history of alcohol consumption. Considering that the amount of khat chewed per session, moderate khat chewers (1 bundle during one chewing session) were shown to be at high risk (AOR = 3.68(1.69–8.01)) and heavy khat chewers had even higher risk (AOR = 18.18(3.56–92.89)) of developing the disease. The odds of developing the disease in those who had no or mild intensity of physical activity were three times higher than in the control group (AOR = 3.01(1.66–5.47)). In those study, participants who used solidified palm oil for food preparation had two time increased odds of developing the diseases (AOR = 2.81(1.49–5.28)) as compared to controls. Also, having dyslipidemia (total cholesterol to HDL ratio >5) increased the chance by six times (AOR = 6.68(2.93–15.23)) as compared to controls Table 3.

## Discussion

The study aimed to assess heavy khat chewing and dyslipidemia as a modifiable risk factor among hypertensive patients in Southwest Ethiopia. This study found educational status (no

**Table 2. Behavioral, nutritional and biochemical hypertensive risk factors among patients in southwest, Ethiopia, 2020.**

| Variable | Categories | Cases No (%) | Controls No (%) | Total No (%) | $X^2$ | p-value |
|---|---|---|---|---|---|---|
| History of Smoking | Yes | 38(27.9%) | 40(14.8%) | 78(19.2%) | 10.04 | 0.002 |
| | No | 98(72.1%) | 230(85.2%) | 328(80.8%) | | |
| Frequency of smoking | Daily | 16(41.0%) | 13(32.5%) | 29(36.7%) | 1.66 | 0.65 |
| | Three times a week | 15(38.5%) | 16(40.0%) | 31(39.2%) | | |
| | Once a week | 6(15.4%) | 10(25.0%) | 16(20.3%) | | |
| | One a month | 2(5.1%) | 1(2.5%) | 3(3.8%) | | |
| Current smoking status | Still smoking | 2(5.3%) | 1(2.5%) | 3(3.8%) | 0.62 | 0.74 |
| | Reduced | 8(21.1%) | 7(17.5%) | 15(19.2%) | | |
| | Ceased | 28(73.7%) | 32(80.0%) | 60(76.9%) | | |
| History of alcohol consumption | Yes | 77(56.6%) | 71(26.3%) | 148(36.5%) | 35.89 | <0.001 |
| | No | 59(43.4%) | 199(73.7%) | 258(63.5%) | | |
| Frequency of Alcohol consumption | Daily | 8(10.4%) | 5(7.0%) | 13(8.8%) | 2.58 | 0.46 |
| | Three times a week | 28(36.4%) | 33(46.5%) | 61(41.2%) | | |
| | Once a week | 18(23.4%) | 11(15.5%) | 29(19.6%) | | |
| | One a month | 23(29.9%) | 22(31.0%) | 45(30.4%) | | |
| Current alcohol status | Still drinking | 25(32.5) | 27(38.0) | 52(35.1%) | 0.50 | 0.78 |
| | Reduced | 18(23.4%) | 15(21.1%) | 33(22.3%) | | |
| | Ceased | 34(44.2%) | 29(40.8%) | 63(42.6%) | | |
| History of Khat Chewing | Yes | 96(70.6%) | 118(43.7%) | 214(52.7%) | 26.23 | <0.001 |
| | No | 40(29.4%) | 152(56.3%) | 192(47.3%) | | |
| Frequency of khat chewing | Daily | 37(38.5%) | 43(36.4%) | 80(37.4%) | 1.41 | 0.70 |
| | Three times a week | 26(27.1%) | 40(33.9%) | 66(30.8%) | | |
| | Once a week | 25(26.0%) | 28(23.7%) | 53(24.8%) | | |
| | Once a month | 8(8.3%) | 7(5.9%) | 15(7.0%0 | | |
| Amount of khat chewed per session | Mild chewer | 30(22.1%) | 79(29.3%) | 109(26.8%) | 59.01 | <0.001 |
| | Moderate chewing | 53(39.0%) | 36(13.3%) | 89(21.9%) | | |
| | Heavy chewing | 13(9.6%) | 3(1.1%) | 16(3.9%) | | |
| | Non-chewers | 40(29.4%) | 152(56.3%) | 192(47.3%) | | |
| Intensity of Physical activity | Mild | 43(31.6%) | 168(62.2%) | 211(52.0%) | 33.94 | <0.001 |
| | Moderate & vigorous- | 93(68.4%) | 102(37.8%) | 195(48.0%) | | |
| Fruit and vegetable intake | <3times/week | 76(55.9%) | 137(51.1%) | 213(52.7%) | 0.82 | 0.37 |
| | >3times/week | 60(44.1%) | 131(48.9%) | 191(47.3%) | | |
| Type of oil for food preparation | Packed(Palm) oil | 105(77.2%) | 172(63.7%) | 277(68.2%) | 7.61 | 0.006 |
| | Sun-flower oil | 31(22.8%) | 98(36.3%) | 129(31.8%) | | |
| Salt you consumption | Optimal | 83(61.0%) | 187(69.3%) | 270(66.5%) | | |
| | High | 53(39.0%) | 83(30.7%) | 136(33.5%) | 2.75 | 0.09 |
| Total Cholesterol | >200mg/dl | 79(58.1%) | 91(33.7%) | 170(41.9%) | 22.09 | <0.001 |
| | <200mg/dl | 57(41.9%) | 179(66.3%) | 236(58.1%) | | |
| HDL level | <40mg/dl | 53(39.0%) | 151(55.9%) | 204(50.2%) | 10.40 | 0.001 |
| | >40mg/dl | 83(61.0%) | 119(44.1%) | 202(49.8%) | | |
| LDL level | >100mg/dl | 103(75.7%) | 147(54.4%) | 250(61.6%) | 17.33 | <0.001 |
| | <100mg/dl | 33(24.3%) | 123(45.6%) | 156(38.4%) | | |
| Triglycerides | >150mg/dl | 73(53.7%) | 91(33.7%) | 164(40.4%) | 14.99 | <0.001 |
| | <150mg/dl | 63(46.3%) | 179(66.3%) | 242(59.6%) | | |
| TC/HDL ratio | >5 | 98(72.1%) | 64(23.7%) | 162(39.9%) | 88.19 | <0.001 |
| | <5 | 38(27.9%) | 206(76.3%) | 244(60.1%) | | |

*(Continued)*

**Table 2.** (Continued)

| Variable | Categories | Cases No (%) | Controls No (%) | Total No (%) | $X^2$ | *p-value* |
|---|---|---|---|---|---|---|
| BMI | Underweight | 37(27.2%) | 48(17.8%) | 85(20.9%) | 5.75 | 0.12 |
| | Normal | 75(55.1%) | 158(58.5%) | 233(57.4%) | | |
| | Over weight | 18(13.2%) | 51(18.9%) | 69(17.0%) | | |
| | Obese | 6(4.4%) | 13(4.8%) | 19(4.7%) | | |

Note: $x^2$, Chi square; BMI, Body mass index.

**Table 3. Bivariate and multivariate logistic regression of heavy khat (Catha edulis) chewing and dyslipidemia as modifiable risk factors among patients in South-west, Ethiopia, 2020.**

| Variable | | Case No (%) | Control No (%) | COR (95%CI) | AOR (95% CI) |
|---|---|---|---|---|---|
| **Educational status** | No formal education | 34(25.0%) | 61(22.6%) | 2.76(1.62–4.70) | **2.25(1.05–4.82)**\* |
| | Primary education | 78(57.4%) | 113(41.9%) | 2.23(1.21–4.12) | 1.45(0.61–3.47) |
| | Secondary and above | 24(17.6%) | 96(35.6%) | 1 | 1 |
| **Income (1dollar = 40birr)** | Low | 78(57.4%) | 132(48.9%) | 1.56(0.92–2.63) | 2.19(0.89–5.42) |
| | Middle | 32(23.5%) | 66(24.4%) | 1.29(0.70–2.38) | 1.74(0.54–5.63) |
| | High | 26(19.1%) | 72(26.7%) | 1 | 1 |
| **History of Smoking** | Yes | 38(27.9%) | 40(14.8%) | 2.23(1.35–3.69) | 1.27(0.56–2.88) |
| | No | 98(72.1%) | 230(85.2%) | 1 | 1 |
| **History of alcohol consumption** | Yes | 77(56.6%) | 71(26.3%) | 3.66(2.37–5.65) | **5.93(3.11–11.30)**\* |
| | No | 59(43.4%) | 199(73.7%) | 1 | 1 |
| **History of Khat Chewing** | Yes | 96(70.6%) | 118(43.7%) | 3.09(1.99–4.80) | 1.55(0.74–3.24) |
| | No | 40(29.4%) | 152(56.3%) | 1 | 1 |
| **Amount of khat chewed per session** | Mild chewer | 29(21.3%) | 79(29.3%) | 1 | 1 |
| | Moderate chewing | 51(37.5%) | 36(13.3%) | 3.88(2.14–7.04) | **3.68(1.69–8.01)**\* |
| | Heavy chewing | 16(11.8%) | 3(1.1%) | 11.41(3.04–42.88) | **18.18(3.56–92.89)**\* |
| | Non-chewers | 40(29.4%) | 152(56.3%) | 0.69(0.40–1.19) | - |
| **Intensity of Physical activity** | Mild | 43(31.6%) | 168(62.2%) | 3.56(2.30–5.52) | **3.01(1.66–5.47)**\* |
| | Moderate/vigorous | 93(68.4%) | 102(37.8%) | 1 | 1 |
| **Type of oil for food preparation** | Packed(Palm) oil | 105(77.2%) | 172(63.7%) | 1.93(1.21–3.09) | **2.81(1.49–5.28)**\* |
| | Sun-flower oil | 31(22.8%) | 98(36.3%) | 1 | 1 |
| **Salt you consumption** | Optimal | 83(61.0%) | 187(69.3%) | 1 | 1 |
| | High | 53(39.0%) | 83(30.7%) | 1.44(0.94–2.21) | 1.30(0.77–2.21) |
| **Total Cholesterol** | >200mg/dl | 79(58.1%) | 91(33.7%) | 2.73(1.78–4.17) | 1.55(0.47–5.12) |
| | <200mg/dl | 57(41.9%) | 179(66.3%) | 1 | 1 |
| **HDL level** | <40mg/dl | 53(39.0%) | 151(55.9%) | 1.98(1.31–3.03) | 1.32(0.74–2.34) |
| | >40mg/dl | 83(61.0%) | 119(44.1%) | 1 | 1 |
| **LDL level** | >100mg/dl | 103(75.7%) | 147(54.4%) | 2.61(1.65–4.14) | 1.26(0.61–2.59) |
| | <100mg/dl | 33(24.3%) | 123(45.6%) | 1 | |
| **Triglycerides** | >150mg/dl | 73(53.7%) | 91(33.7%) | 2.28(1.49–3.47) | 1.59(0.51–4.96) |
| | <150mg/dl | 63(46.3%) | 179(66.3%) | 1 | |
| **TC/HDL ratio** | >5 | 98(72.1%) | 64(23.7%) | 8.30(5.19–13.25) | **6.68(2.93–15.23)**\* |
| | <5 | 38(27.9%) | 206(76.3%) | 1 | |

Note: \*, p<0.05.

formal education), excess alcohol consumption, the amount of bundles of khat chewed per session; the intensity of physical activity, type of oil used for food preparation, and dyslipidemia as the risk factors for the development of the disease.

Educational status affects the development of hypertension. Those who had no formal education were 2.25 times more likely to be exposed to hypertension. This was in line with a study in Gaza Governorates where the risk of developing hypertension was five times higher among low educational level [19]. In contrast to this study, a study in Bale, Nekemte Ethiopia, and Korea showed, having a secondary and above educational level increases the chance of the development of hypertension [12, 20, 29]. The variation observed may be explained by the differences in methodological approach, socio-economic status of the study population, cultural, dietary and lifestyle differences of the areas and countries.

Moderate khat chewers (1 bundle during one chewing session) and heavy khat chewers (>2 bundles during one chewing session) had a higher risk of developing a disease. This was consistent with a study in Bahirdar, Ethiopia that associates khat chewing with elevated systolic blood pressure [28]. A study in Butajira, Ethiopia showed regular khat chewing increased diastolic blood pressure [10]. Another observational study in Yemeni khat chewers who had acute coronary syndrome showed that khat chewers had worse outcomes than non-chewers [14]. But in contrast, another study in Ethiopia found there is no association [12]. This may support the fact that khat chewing is a powerful chronic illness determinant and reducing or cessation is the single most effective lifestyle measure to prevent numerous deaths and disabilities related to hypertension. A study on healthy volunteers investigated that khat chewing leads to a significant and progressive rise in systolic and diastolic blood pressure and heart rate [30]. And for the difference, there needs to be a more experimental study that should be carried on for the future.

In this study, community obesity is not a problem; there is no association between anthropometric measurements and being hypertensive. This might be due to the high consumption of khat in the study population and the community because khat chewing results in anorexia, which suppresses appetite [6].

Several researchers reported sedentary lifestyle is related to hypertension [31, 32]. Also in this study, those who participate in mild intensity physical activity had three times increased odds of developing the disease as compared to moderate and vigorous-intensity physical activity. This study is in line with different studies [2, 33, 34]. Many epidemiological studies showed that repeated physical activity results in significant BP and weight reduction. Sedentary life, which is a predictor of obesity, is one of the main risk factors of high blood pressure [35, 36]. But another study in Tigray, north Ethiopia related vigorous work-related activity as a risk factor [17]. This might be because of stressful activities increasing stress in everyday life activity and the release of stress hormones. But to compare to those who have a sedentary lifestyle, moderate physical activity is protective but different studies were against this [19, 28].

The odd of developing hypertension in those with history of excess alcohol consumption was 5.93 times more likely than the control groups. Because alcohol consumption increases the odds of hypertension due to its sympathetic effect and in line with other studies [37–39].

In our finding, using a solidified palm oil is another risk factor where the odds of having hypertension were increased by two times than the control groups. A study in Assosa Ethiopia showed that the level of total saturated fatty acid ranged from 14.3% for sunflower oil to 69.97% for chief palm oil, with the predominant presence of palmitic acid and stearic acid. Especially total saturated fatty acid in Chief palm oil accounted for more than 2/3 of the total fatty acid. The high saturated fatty acid content of Chief palm oil differed from other reports [40]. Another study in Gonder, Ethiopia which studied a risk factor for metabolic syndrome supported this [41–43].

Our finding showed a significant association between hypertension and abnormal lipid profile where the odds of developing hypertension were increased by 6.68 times compared to control group. Dyslipidemia appeared to be the strongest predictor in determining the probability of having hypertension [44–46]. A study in north Ethiopia showed that abnormal lipid profile especially having low HDL-c being the most commonly encountered abnormality [47]. Another study among Japanese men showed that elevated serum levels of TC/HDL ratio increased the risk of hypertension [48]. This is because dyslipidemia causes endothelial damage and the loss of physiological vasomotor activity those results from endothelial damage which is manifested as increased blood pressure (BP).

Though institution based cases were selected to minimize selection bias, as a result of the nature of case-control study the temporal relationships of events between explanatory variables and hypertension cannot be determined. And also cases were selected consecutively as soon as they were identified, selection bias might be introduced. Recall bias and social desirability bias were also limitations that might have affected the accuracy of information as the respondents were asked questions about some of their previous health-related events. Moreover, the findings of the current research cannot be generalized to the whole community, because of its institution-based nature.

## Conclusion

The study showed that modifiable risk factors were the main risk factor for the development of hypertension. The findings of this study highlighted that health education is needed to focus on physical exercise, quitting excess alcohol consumption, quitting khat chewing by giving special emphasis to those who had no formal education and also focusing on a healthy diet free of cholesterol. We recommend that the policymakers need to focus on community-level intervention through integration to health extension programs. It is also better to give special emphasis to health education regarding a healthy diet.

## Supporting information

**S1 Questionnaire. English version questionnaires.**
(DOCX)

**S2 Questionnaire. Amharic version questionnaires.**
(DOCX)

**S3 Questionnaire. Afaan Oromoo version questionnaires.**
(DOCX)

**S1 Data. The SPSS data.**
(SAV)

## Acknowledgments

We would like to thank Jimma University and thank with the deepest gratitude all participants who took part in this study for their commitment to give valuable information and cooperation.

## Author Contributions

**Conceptualization:** Meron Hadis Gebremedhin, Lielt Gebreselassie Gebrekirstos.

**Data curation:** Meron Hadis Gebremedhin, Lielt Gebreselassie Gebrekirstos.

**Formal analysis:** Meron Hadis Gebremedhin, Lielt Gebreselassie Gebrekirstos.

**Funding acquisition:** Meron Hadis Gebremedhin.

**Investigation:** Meron Hadis Gebremedhin.

**Methodology:** Meron Hadis Gebremedhin, Eyasu Alem Lake, Lielt Gebreselassie Gebrekirstos.

**Project administration:** Meron Hadis Gebremedhin.

**Resources:** Meron Hadis Gebremedhin, Eyasu Alem Lake.

**Software:** Meron Hadis Gebremedhin.

**Supervision:** Meron Hadis Gebremedhin, Eyasu Alem Lake.

**Validation:** Meron Hadis Gebremedhin, Lielt Gebreselassie Gebrekirstos.

**Writing – original draft:** Meron Hadis Gebremedhin.

**Writing – review & editing:** Meron Hadis Gebremedhin, Eyasu Alem Lake, Lielt Gebreselassie Gebrekirstos.

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
