## [Decision Letter · Decision Letter 0]

10 Mar 2021

PONE-D-20-40692

Heavy khat (Catha edulis) chewing and dyslipidemia as a cardiovascular risk factor among hypertensive patients in Jimma, Ethiopia: Unmatched case control study

PLOS ONE

Dear Dr. Gebremedhin,

Thank you for submitting your manuscript to PLOS ONE. After careful consideration, we feel that it has merit but does not fully meet PLOS ONE’s publication criteria as it currently stands. Therefore, we invite you to submit a revised version of the manuscript that addresses the points raised during the review process. The reviewers have raised critical concerns on the sample size estimation, measurement of key variables and data analysis that you need to be address. Further, the organization and write up of the manuscript have to be improved. 

We look forward to receiving your revised manuscript.

Kind regards,

Samson Gebremedhin, PhD

Academic Editor

PLOS ONE

Journal Requirements:

2. Please include additional information regarding the survey or questionnaire used in the study and ensure that you have provided sufficient details that others could replicate the analyses. For instance, if you developed a questionnaire as part of this study and it is not under a copyright more restrictive than CC-BY, please include a copy, in both the original language and English, as Supporting Information. Moreover, please include more details on how the questionnaire was pre-tested, and whether it was validated.

3.We note that you have indicated that data from this study are available upon request. PLOS only allows data to be available upon request if there are legal or ethical restrictions on sharing data publicly. For information on unacceptable data access restrictions, please see http://journals.plos.org/plosone/s/data-availability#loc-unacceptable-data-access-restrictions.

4.We note that the grant information you provided in the ‘Funding Information’ and ‘Financial Disclosure’ sections do not match.

Reviewers' comments:

Reviewer's Responses to Questions

**Comments to the Author**

1. Is the manuscript technically sound, and do the data support the conclusions?

Reviewer #1: No

Reviewer #2: Partly

2. Has the statistical analysis been performed appropriately and rigorously? 

Reviewer #1: Yes

Reviewer #2: No

3. Have the authors made all data underlying the findings in their manuscript fully available?

Reviewer #1: No

Reviewer #2: No

4. Is the manuscript presented in an intelligible fashion and written in standard English?

Reviewer #1: No

Reviewer #2: No

5. Review Comments to the Author

Reviewer #1: Dear editor,

Thank you for inviting me to review this manuscript. I read the manuscript with much interest, especially given that khat is purported to be a risk factor for cardiovascular disorders and has been less subjected to epidemiological studies in Ethiopia. The present study has shown, using a case-control study, that hypertension is associated with khat chewing and other factors such as dyslipidemia. The paper is potentially publishable; however, as it stands now it doesn’t seem suitable for publication. The author needs to do a very good job of revising the manuscript if it is to be acceptable for publication.

I have provided my comments on the manuscript section by section next.

Language:

The manuscript needs to be extensively revised for language (grammar and mechanics) starting from the title. As it currently stands, it is unsuitable for publication.

Title:

The study is not about cardiovascular risk factors among hypertensive patients. It is about risk factors for hypertension. Hence, the title needs to be accordingly revised.

Abstract:

Make sure that the findings presented in the results part of the abstract make sense. For example, is the finding that work-related vigorous physical activity increases the odds of hypertension plausible and logical? The author needs to re-check the analysis. Also, to what level of education does “educational status” refer to? The author should also accompany abbreviations with their expanded form when they first appear in the abstract.

Furthermore, the author should conclude only based on the findings. This study has nothing about whether minimizing khat cultivation by farmers would have an effect in reducing khat consumption and thereby the risk of hypertension. In countries where khat is not cultivated, it may be imported from other countries. Reducing cultivation may not necessarily reduce khat consumption. That conclusion is off-topic. The recommendation about increasing educational attainment also does not seem to be feasible and may not have an effect in the short term. The recommendations about stress and type of food oil are also not supported by the findings of the present study.

Introduction:

The arguments in the introduction lack coherence. Make sure that you coherently address one main idea in a paragraph. As the introduction currently stands, each paragraph seems to entertain multiple unrelated ideas and hence makes reading through it difficult.

Accompany all abbreviations with their full form when they appear first in the body of the manuscript.

In the last paragraph of the introduction, the argument about the objectives of the present study is confusing. Is the aim of the study to investigate the association between khat chewing and dyslipidemia or the association of khat chewing and dyslipidemia with hypertension? You should clearly re-write this argument.

Methods and Materials:

Reference #20 does not seem a proper reference for the information presented about the study setting. For one thing, it is out-of-date. Besides, it is not an original source for such information.

Description of the study population should be revised. It doesn't sound right. On the one hand, all hypertensive patients do not seem to be in the study population. They were recruited through a sampling procedure based on a sample size calculated a-priori. On the other hand, the study is not limited solely to hypertensive patients.

While defining cases on page 5, the author should clarify based on how many occasions of blood pressure measurement hypertension was defined.

When describing the methodology, do not do it in the first person singular. Eg., "I selected two controls for each case." (Page 5, last line.)

Description of the inclusion criteria for both the cases and the controls on page 6 must be properly revised. For one thing, the description is unnecessarily long. Besides, the criteria should be clear and properly justified. For example, once you stated that the population studied was comprised of adults > 18 years, then no need of stating that persons under 18 years were excluded.

The description of the sample size calculation on page 6 is unclear. The language is awkward and the assumptions are not clearly stated. For example, what was the exposure variable? You also need to cite reference for the assumptions about the magnitude of exposure among the cases and the controls. Further, what do you mean by the "standard sample size formula"? If you used software to calculate the sample size, better to omit the description about the formula.

Merge the description of the sampling procedure with the description of the sample size. Further, the description of the sampling procedure is too brief and too vague. Clear and sufficient description of the sampling procedure both for the cases and the controls should be provided.

In the data collection section (page 7), write the proper name of the local language - Afaan Oromoo. Besides, the description about the definition of hypertension is unclear. Also the description of hypertensive patients on medication is confusing. Should be re-written.

Remove the operational definitions section (pages 8 & 9) and provide a description of how the different variables were measured and operationalized under the variables of the study section (page 8). Further, the main independent variables, namely heavy khat use and dyslipidemia need explicit elaboration. As the description stands now, it is difficult to properly understand how the IDVs were measured and operationalized. As necessary, sources for the given definitions need to be provided by way of reference citation.

Assimilate the data quality management section into the data collection procedure section.

Results:

Under the sociodemographic characteristics section, describe only the salient sociodemographic features in text. The rest can be seen from the table. Also clearly indicate in terms of which characteristics the cases and controls significantly differ.

In Table 1, do a statistical comparison of the cases and controls to show whether or not they were balanced in terms of basic characteristics. You may use chi-square test.

The descriptive results about the biochemical, behavioural and nutritional characteristics of the study participants (pages 13-14) are unclear and confusing. Just provide a brief description of the salient aspects of these findings under one sub-heading titled "biochemical, behavioural and nutritional characteristics". Like in Table 1, here also provide in a table the statistical comparison of the cases and the controls in terms of characteristics provided. Replace Table 2 with a table similar to the one I suggested for Table 1.

The study is specifically about risk factors for hypertension. Hence you should modify the sub-heading "Cardiovascular risk factors" on page 19 with a more descriptive sub-heading such as "Risk factors for hypertension".

Avoid discussion from the results section. For example, the arguments about choice of analytical methods provided on page 19 may be justified in the methods section, not in the results. Further, such arguments must be substantiated by appropriate reference citations to convince readers about the trustworthiness of the claims.

In the methods section, nothing was described about the use of unconditional logistic regression but in the results (page 19) there is a claim of the use of such method. If unconditional logistic regression was used, then in the methods section the use of this method should be clearly described.

As heavy khat use has not been clearly defined in the methods section, it is difficult to understand the result pertaining to the association of heavy khat use and hypertension on page 19.

Further, as already commented in the abstract, the finding that vigorous work-related activity increases the odds of hypertension is not plausible. The authors should re-do the analysis using an appropriate reference category and check the findings.

In Table 3, the columns for AOR and COR should be transposed. Also better to omit the p-value columns.

In Table 3, is the number of bundles chewed per day or per session or per what?

Discussion:

The author should revise the discussion to make it more mature and scholarly.

Remove the introductory statement about khat from the first paragraph of the discussion. In the first paragraph, just summarize the main findings and discuss those findings in subsequent paragraphs.

Do not sensationalize the discussion of the findings (e.g., “But to surprise”, line 310, page 23).

Your study did not investigate the association of khat chewing with dyslipidemia and HDL. Your argument on page 23 (line 310-311) is not supported by your data and is misleading.

Further, the result pertaining to the association between dyslipidemia and hypertension needs to be sufficiently discussed.

As already commented above, the claim on the last paragraph of page 23 that work-related vigorous physical activity increases the odds of hypertension is counterfactual and not plausible. You need to re-check your analysis and update the discussion accordingly.

The first sentence in the last paragraph of the discussion (page 24) that the present study is the first study in the study setting seems an exaggeration of the findings. There may already be a study which the author failed to find. At least, it is already known that dyslipidemia is a risk factor for hypertension. Revise or avoid such claim.

Clearly discuss the limitations of this study.

Conclusion:

The conclusion seems to stray away from the objective of the study and from what the data shows. For example, the first sentence of the conclusion section is wrong and not in line with the objective of the study. The second sentence is also unclear and not in line with the study's objective. Generally, the author should revise the conclusion to be in line with the objective of the study and suggest a pertinent recommendation about what should be done.

Reviewer #2: The study is relevant as it tries to assess the factors associated with hypertension in Ethiopia. Knowing these factors will contribute to the implementation of policies and adoption of lifestyle aimed at combating hypertension. However, there are several major concerns or issues that must be addressed to make the manuscript suitable for publication.

Major issues:

1. The title is not correct. I should read “Heavy khat (Catha edulis) chewing and dyslipidemia as cardiovascular risk factors among hypertensive patients in Jimma, Ethiopia: Unmatched case3 control study”

2. On page 8, under biochemical analysis you did describe the quality control processes that were undertaken to ensure the reliability of the results. How did you measure these results? Did you use an automated analyzer and what was the name of the analyzer and the country or company of manufacture?

3. You have not stated how your independent variables were collected or have not defined them.

4. In your data analysis on page 9, you did not state which variables were controlled for. Similarly, on page 19, line 256, you stated that possible confounders were adjusted for. What were these confounders?

5. On page 19, line 257, you mentioned that excess alcohol consumption was associated with hypertension. The bivariate analysis in the table did not mention excess alcohol consumption. The category there is “still drinking” and this is different from excess alcohol consumption.

6. If after adjusting for confounders excess alcohol consumption, educational status, vigorous work-related activity etc. in addition to Khat chewing were each independently associated with hypertension why is your focus only on Khat chewing? If your interest ids on Khat chewing then you need to adjust for all other variables (confounders) that are associated hypertension before you determine whether Khat chewing is independently associated with hypertension.

7. Again, in your conclusion you stated other factors that are associated with hypertension. Why should Khat chewing be given particular attention.

Minor issues:

1. On page 7, state the reference/source for the cut off value for hypertension using the National Committee Categorization of Hypertension.

2. Define OCP in both tables 1 and 2 and COR in table 2.

3. The statement from lines 73 to 75 under the introduction is not correct. Check this “…..and the association of the two variables and other factors to the study area and our

75 country, Ethiopia”

4. The statement in line 46 and 47 is quite ambiguous. Which percentage is for which condition?

6. PLOS authors have the option to publish the peer review history of their article (what does this mean?). If published, this will include your full peer review and any attached files.

Reviewer #1: **Yes: **Ayalew Astatkie

Reviewer #2: No

---

## [Author Response · Author response to Decision Letter 0]

18 Jun 2021

Dear editor and reviewers, I have included a response for all the points raised, please see the attached document as "response to reviewers".

---

## [Decision Letter · Decision Letter 1]

11 Jul 2021

PONE-D-20-40692R1

Heavy khat (Catha edulis) chewing and dyslipidemia as modifiable hypertensive risk factors among patients in Southwest, Ethiopia: Unmatched case-control study

PLOS ONE

Dear Dr. Gebremedhin,

Thank you for submitting your manuscript to PLOS ONE. After careful consideration, we feel that it has merit but does not fully meet PLOS ONE’s publication criteria as it currently stands. Therefore, we invite you to submit a revised version of the manuscript that addresses the points raised during the review process. Please address all the concerns raised by the reviewer regards measurement of key variables, data analysis and write of the manuscript. Further, please provide justification why you added new authors and why it was not possible to include them in the first submission. 

We look forward to receiving your revised manuscript.

Kind regards,

Samson Gebremedhin, PhD

Academic Editor

PLOS ONE

Reviewers' comments:

Reviewer's Responses to Questions

**Comments to the Author**

1. If the authors have adequately addressed your comments raised in a previous round of review and you feel that this manuscript is now acceptable for publication, you may indicate that here to bypass the “Comments to the Author” section, enter your conflict of interest statement in the “Confidential to Editor” section, and submit your "Accept" recommendation.

Reviewer #1: (No Response)

2. Is the manuscript technically sound, and do the data support the conclusions?

Reviewer #1: Partly

3. Has the statistical analysis been performed appropriately and rigorously? 

Reviewer #1: No

4. Have the authors made all data underlying the findings in their manuscript fully available?

Reviewer #1: Yes

5. Is the manuscript presented in an intelligible fashion and written in standard English?

Reviewer #1: No

6. Review Comments to the Author

Reviewer #1: Dear editor,

Thank you for inviting me to review the revised version (Revision 1) of this manuscript. I would like to witness that the manuscript has been substantially improved relative to the initial submission. However, the authors still need to do some more work in order to make the manuscript suitable for publication.

I have provided my comments and suggestions below.

The initial submission had only a single author. However the present submission has two additional authors (totally three). The submitting author should justify if these newly included authors really deserve being listed as authors.

The entire manuscript still needs careful language revision. As it stands now, the language is not suitable for publication.

The description in the third paragraph of the introduction (lines 65-70) is confusing and not coherent. It needs to be re-written.

The fourth paragraph of the introduction (lines 71-79) doesn't connect smoothly to the preceding and succeeding paragraphs. Must be revised.

In the description of the study setting, line 97, provide the URL of JUMC.

The main exposure variables of interest in the present study were "heavy khat use" and "dyslipidemia". So why was "past history of cigarette smoking" used as an exposure variable in the sample size calculation? It is not in line with the objective of the study.

The sampling procedure still needs a clearer description.

In the initial submission, it was stated that the tool was translated to the local language in Oromia (i.e., Afaan Oromoo). In the revised submission, the authors claim that the tool was translated to Amharic, and there is no mention of the language stated in the initial submission. This casts doubt on the trustworthiness of the authors’ claims.

The reason for back-translation should be revised. The reason stated as “to check for internal consistency” is not sound and convincing.

On page 9, the authors have provided a detailed procedure for arterial blood pressure measurement. However, earlier in the manuscript, they have stated that "Cases were patients who were already diagnosed with hypertension by a physician or those taking antihypertensive drugs during the study period." If that is how cases were ascertained, how did the authors make sure that the BP measurement for cases was done as per the procedure described there? Clearly, measurement of blood pressure for cases was not under the control of the authors.

The reference cited in relation to the use of a p-value of 0.25 for selection of variables for the multivariable model (reference #21, page 11) does not seem to be the appropriate reference for the argument provided. The cited reference (#21) is a book about clinical examination and has nothing to do with statistical data analysis. The appropriate reference must be cited. Also check the appropriateness of all other citations.

Better to describe the results pertaining to physical activity in terms of the intensity of the physical activity. The “active/inactive” dichotomy doesn’t seem to make much sense.

One of the independent variables used in the multivariable analysis was oral contraceptive use. However, oral contraceptive use applies only to females. Such a variable which applies to only a sub-set of the sample should not be used in multivariable analysis as it will significantly diminish the total sample used in the multivariable analysis. In the present manuscript, including OC use will preclude all males from the analysis.

Further, the analysis of "amount of khat chewed per session" should include “non-chewers” in the analysis so that the analysis will apply to the entire sample, not to a sub-set of the sample (khat chewers) only. Hence, recode the "amount of khat chewed per session" as "non-chewer, mild khat chewer, moderate khat chewer, and heavy khat chewer" and re-do the analysis.

Do not state odds ratios with 95% CIs in the discussion section.

The results in the present study pertaining to the association of level of education with hypertension are contrary to several previous studies. The authors tried to justify this simply as “…because of the difference in the socio-economic status of the study population, cultural, dietary and lifestyle differences of the areas and countries,” (lines 287-289). What if it is due to methodological problem with the present study? The authors should re-check the analysis. Even if the re-analysis may not change the results, the argument the authors come up with must be very critical and convincing.

In lines 299-300, the authors recommend an experimental study to further investigate the association of khat chewing with hypertension-related morbidity and mortality. Is that practicable and ethically sound?

Discussion of the effect of physical activity on hypertension risk (lines 304-310) must be done carefully and critically taking account of the whole body of evidence on the topic. The authors' arguments in their present form lack a critical analysis of the state-of-the-art evidence.

In the last paragraph of the discussion, the authors need to provide careful and critical discussion of the limitations of the present study. What has been provided is very superficial and not convincing.

Check the authors' guideline of PLoS ONE for the appropriate placement of tables within the manuscript.

In Tables 1 & 2, include a p-value column to show whether the differences were significant or not.

Better to remove the p-values from Table 3 as it is unclear whether they were based on the crude analysis or the adjusted analysis. Besides, ORs with 95% CIs can tell both about the magnitude and significance of the association. Hence, the use of a p-value is superfluous. Also, include columns which provide the cross-tabulation of each independent variable with the dependent variable in Table 3.

7. PLOS authors have the option to publish the peer review history of their article (what does this mean?). If published, this will include your full peer review and any attached files.

Reviewer #1: **Yes: **Ayalew Astatkie

---

## [Author Response · Author response to Decision Letter 1]

8 Sep 2021

Manuscript Title: Heavy khat (Catha edulis) chewing and dyslipidemia as modifiable hypertensive risk factors among patients in Southwest, Ethiopia: Unmatched case-control study

Dear editor and reviewers 

We would like to extend our heartfelt gratitude and appreciation for your valuable comments and priceless time. Thank you, your comments have helped us a lot to improve the manuscript and focus on important points and also a base for our future performance. We are very happy with all the points raised because this is not only a comment rather we learnt a lot. We tried to address all comments point by point in this paper. 

Responses for the editor 

1. Dear editor, we tried to include all concerns raised by the reviewer in the response to the reviewer section below including measurement of key variables, data analysis, and write of the manuscript and justification for adding new authors.

Response to reviewer 1

1. The initial submission had only a single author. However the present submission has two additional authors (totally three). The submitting author should justify if these newly included authors really deserve being listed as authors.

Response: Thanks for the comment and sorry for not including them in the initial submission. Of course this authors participated in the work and there specific role has been listed in the “author contribution”. A vast study had been carried out on some chronic illness and some of the data is under write up and this paper concerning hypertension is taken from it. 

2. The entire manuscript still needs careful language revision. As it stands now, the language is not suitable for publication.

Response: Comment accepted and we tried to make changes. 

3. The description in the third paragraph of the introduction (lines 65-70) is confusing and not coherent. It needs to be re-written. 

Response: Comment accepted and amendments have been done. Please see the highlighted manuscript from line (62-85).

4. The fourth paragraph of the introduction (lines 71-79) doesn't connect smoothly to the preceding and succeeding paragraphs. Must be revised.

Response: Comment accepted and changes have been made. Please see the highlighted manuscript again from line (62-85).

5. In the description of the study setting, line 97, provide the URL of JUMC.

Response: Comment accepted and the URL of JUMC was provided. Please see the highlighted manuscript from line (94).

6. The main exposure variables of interest in the present study were "heavy khat use" and "dyslipidemia". So why was "past history of cigarette smoking" used as an exposure variable in the sample size calculation? It is not in line with the objective of the study.

Response: The sample size was calculated and fixed during the development of proposal. And the title was modified after analysis because heavy khat chewing and dyslipidemia have a higher odds ratio. Any further recommendation on the title can be accepted. 

7. The sampling procedure still needs a clearer description.

Response: Comment accepted and changes have been made. Please see the highlighted manuscript from line (136-140).

8. In the initial submission, it was stated that the tool was translated to the local language in Oromia (i.e., Afaan Oromoo). In the revised submission, the authors claim that the tool was translated to Amharic, and there is no mention of the language stated in the initial submission. This casts doubt on the trustworthiness of the authors’ claims.

Response: Comment accepted and changes have been made and we will attach questionnaires with local languages. 

9. The reason for back-translation should be revised. The reason stated as “to check for internal consistency” is not sound and convincing.

Response: Comment accepted and changes have been made. Please see the highlighted manuscript from line (149-151).

10. When On page 9, the authors have provided a detailed procedure for arterial blood pressure measurement. However, earlier in the manuscript, they have stated that "Cases were patients who were already diagnosed with hypertension by a physician or those taking antihypertensive drugs during the study period." If that is how cases were ascertained, how did the authors make sure that the BP measurement for cases was done as per the procedure described there? Clearly, measurement of blood pressure for cases was not under the control of the authors.

Response: Comment accepted and data collectors were taken from the chronic illness clinic where the same procedure was carried out. And the procedure was used for the control groups, too.

11. The reference cited in relation to the use of a p-value of 0.25 for selection of variables for the multivariable model (reference #21, page 11) does not seem to be the appropriate reference for the argument provided. The cited reference (#21) is a book 1about clinical examination and has nothing to do with statistical data analysis. The appropriate reference must be cited. Also check the appropriateness of all other citations.

Response: Comment accepted and changes have been made. Please see the highlighted manuscript from line (204).

12. Better to describe the results pertaining to physical activity in terms of the intensity of the physical activity. The “active/inactive” dichotomy doesn’t seem to make much sense.

Response: Comment accepted and modifications have been made. Please see the highlighted manuscript from line (235-236) and Table 2 and 3.

13. One of the independent variables used in the multivariable analysis was oral contraceptive use. However, oral contraceptive use applies only to females. Such a variable which applies to only a sub-set of the sample should not be used in multivariable analysis as it will significantly diminish the total sample used in the multivariable analysis. In the present manuscript, including OC use will preclude all males from the analysis.

Response: Comment accepted and amendment has been done. We tried to screen starting from Epi data and coding in SPSS. There was of course some gap with the analysis and even coding. But we did rigorous analysis to include all raised concerns. Please see the Table 2 and Table 3. 

14. Further, the analysis of "amount of khat chewed per session" should include “non-chewers” in the analysis so that the analysis will apply to the entire sample, not to a sub-set of the sample (khat chewers) only. Hence, recode the "amount of khat chewed per session" as "non-chewer, mild khat chewer, moderate khat chewer, and heavy khat chewer" and re-do the analysis. 

Response: Thanks for the comment and “non-chewer” has been included. Please see Table 2 & Table 3).

15. Do not state odds ratios with 95% CIs in the discussion section.

Response: Comment accepted and amendment has been done. 

16. The results in the present study pertaining to the association of level of education with hypertension are contrary to several previous studies. The authors tried to justify this simply as “…because of the difference in the socio-economic status of the study population, cultural, dietary and lifestyle differences of the areas and countries,” (lines 287-289). What if it is due to methodological problem with the present study? The authors should re-check the analysis. Even if the re-analysis may not change the results, the argument the authors come up with must be very critical and convincing.

Response: Comment accepted and amendment has been done. Please see from line (280-282).

17. In lines 299-300, the authors recommend an experimental study to further investigate the association of khat chewing with hypertension-related morbidity and mortality. Is that practicable and ethically sound?

Response: Thanks for the comment. Though old we had included some literatures regarding the effect of khat on myocardial function and effect on blood vessels. So, further clinical trials involving animal studies can be carried out to strengthen the result (292-295). 

18. Discussion of the effect of physical activity on hypertension risk (lines 304-310) must be done carefully and critically taking account of the whole body of evidence on the topic. The authors' arguments in their present form lack a critical analysis of the state-of-the-art evidence.

Response: Comment accepted and amendment has been done. Please see the highlighted manuscript from line (300-309).

19. In the last paragraph of the discussion, the authors need to provide careful and critical discussion of the limitations of the present study. What has been provided is very superficial and not convincing.

Response: Comment accepted and amendment has been done. Please see the highlighted manuscript from line (330-336).

20. Check the authors' guideline of PLoS ONE for the appropriate placement of tables within the manuscript.

Response: Comment accepted and changes have been made. Please see Table1-3.

21. In Tables 1 & 2, include a p-value column to show whether the differences were significant or not.

Response: Comment accepted and change has been made. Please see Table1&2.

22. Better to remove the p-values from Table 3 as it is unclear whether they were based on the crude analysis or the adjusted analysis. Besides, ORs with 95% CIs can tell both about the magnitude and significance of the association. Hence, the use of a p-value is superfluous. Also, include columns which provide the cross-tabulation of each independent variable with the dependent variable in Table 3.

Response: Comment accepted and changes have been made. Please see Table 3.

---

## [Decision Letter · Decision Letter 2]

27 Sep 2021

PONE-D-20-40692R2Heavy khat (Catha edulis) chewing and dyslipidemia as modifiable hypertensive risk factors among patients in Southwest, Ethiopia: Unmatched case-control studyPLOS ONE

Dear Dr. Gebremedhin,

Thank you for submitting your manuscript to PLOS ONE. After careful consideration, we feel that it has merit but does not fully meet PLOS ONE’s publication criteria as it currently stands. Therefore, we invite you to submit a revised version of the manuscript that addresses the points raised during the review process. Specifically please address the comments raised by the reviewer including those on selection of variables for the multivariable model and formatting, structuring and writeup of the manuscript. 

We look forward to receiving your revised manuscript.

Kind regards,

Samson Gebremedhin, PhD

Academic Editor

PLOS ONE

Additional Editor Comments (if provided):

Please address comments of the reviewers including those related to selection of variables for the multivariable model, structuring and writeup of the manuscript.

Reviewers' comments:

Reviewer's Responses to Questions

**Comments to the Author**

1. If the authors have adequately addressed your comments raised in a previous round of review and you feel that this manuscript is now acceptable for publication, you may indicate that here to bypass the “Comments to the Author” section, enter your conflict of interest statement in the “Confidential to Editor” section, and submit your "Accept" recommendation.

Reviewer #1: (No Response)

2. Is the manuscript technically sound, and do the data support the conclusions?

Reviewer #1: Partly

3. Has the statistical analysis been performed appropriately and rigorously? 

Reviewer #1: No

4. Have the authors made all data underlying the findings in their manuscript fully available?

Reviewer #1: Yes

5. Is the manuscript presented in an intelligible fashion and written in standard English?

Reviewer #1: Yes

6. Review Comments to the Author

Reviewer #1: Many of my previous concerns have been addressed. However, there are still concerns remaining to be addressed. My major concern is the use of OCP use in the multivariable model (see comment 8 below). The authors should give careful attention to that comment.

Hereunder are my remaining concerns.

1. The language still needs careful revision.

2. The term "physical inactivity" in the abstract must be revised.

3. One of the justifications the authors provided for doing the present study in lines 78-80 is "Because previous studies conducted in Ethiopia were cross-sectional in nature which lacked a cause-and-effect relationship." The case-control design used by the present study also has the same limitation. Therefore, that argument is not convincing. It should be omitted or re-phrased.

4. Better to indicate the URL of JUMC in the text describing the study setting instead of giving it a reference number and citing it as a formal reference. If it has to be cited formally, the reference information should comprise of all important information for a Web page used as a reference.

5. In the data collection section, the authors should state into what local languages the questionnaire was translated.

6. The information provided under "Arterial blood pressure measurement" (pages 9-10) should be part of the "Measurement of variables" section (page 7). Hence, merge it with the description of the "Outcome variable". Also avoid bulleted listing. Provide your arguments using complete sentences.

7. The response the authors provided for my earlier comment regarding measurement of blood pressure for cases is not convincing. Merely taking data collectors from the chronic illness clinic does not make the blood pressure measurement for cases similar to the blood pressure measurement of the controls.

8. The authors have still retained OCP use in the multivariable analysis. As I commented in my previous review, OCP use applies only to females and as such should not be used as an independent variable in the multivariable model because its inclusion automatically excludes all males from the multivariable model. As it is seen from the cross tabulation in Table 3, the sample size for USP use is only 193, which is the sample size for female study participants. What the authors should clearly recognize is, in the multivariable model if the sample size for OCP use is 193, then the sample size for all other variables in the model is automatically diminished to 193. That is, 213 study participants (all males) are excluded from the analysis for all independent variables in the model. Therefore, the authors should re-do the analysis by excluding OCP use if the results of the present study are to be valid.

9. The authors should still check the authors' guideline of PLoS ONE for the appropriate placement of tables within the manuscript.

10. The discussion still needs to be more critical.

11. The authors should make sure that all references are properly written.

12. The data availability statement should be in line with PLoS ONE’s data availability policy.

7. PLOS authors have the option to publish the peer review history of their article (what does this mean?). If published, this will include your full peer review and any attached files.

Reviewer #1: **Yes: **Ayalew Astatkie

---

## [Author Response · Author response to Decision Letter 2]

3 Oct 2021

Response to reviewer 1

1. The language still needs careful revision.

Response: Comment accepted and changes had been done. 

2. The term "physical inactivity" in the abstract must be revised.

Response: Comment accepted and amendment has been done. Please see the highlighted manuscript line (32).

3. One of the justifications the authors provided for doing the present study in lines 78-80 is "Because previous studies conducted in Ethiopia were cross-sectional in nature which lacked a cause-and-effect relationship." The case-control design used by the present study also has the same limitation. Therefore, that argument is not convincing. It should be omitted or re-phrased.

Response: Comment accepted and changes have been made. Please see the highlighted manuscript again from line (78-82).

4. Better to indicate the URL of JUMC in the text describing the study setting instead of giving it a reference number and citing it as a formal reference. If it has to be cited formally, the reference information should comprise of all important information for a Web page used as a reference.

Response: Comment accepted and the URL of JUMC was included in the manuscript. Please see the highlighted manuscript line (91).

5. In the data collection section, the authors should state into what local languages the questionnaire was translated.

Response: Comment accepted and changes have been made. Please see the highlighted manuscript from line (158-159).

6. The information provided under "Arterial blood pressure measurement" (pages 9-10) should be part of the "Measurement of variables" section (page 7). Hence, merge it with the description of the "Outcome variable". Also avoid bulleted listing. Provide your arguments using complete sentences.

Response: Comment accepted and changes have been made. Please see the highlighted manuscript from line (112-121).

7. The response the authors provided for my earlier comment regarding measurement of blood pressure for cases is not convincing. Merely taking data collectors from the chronic illness clinic does not make the blood pressure measurement for cases similar to the blood pressure measurement of the controls.

Response: Cases were already selected because they were confirmed hypertensive patients OR on anti-hypertensive treatment and controls were confirmed to be free from hypertension by following the same procedure followed in the chronic illness clinics. 

8. The authors have still retained OCP use in the multivariable analysis. As I commented in my previous review, OCP use applies only to females and as such should not be used as an independent variable in the multivariable model because its inclusion automatically excludes all males from the multivariable model. As it is seen from the cross tabulation in Table 3, the sample size for USP use is only 193, which is the sample size for female study participants. What the authors should clearly recognize is, in the multivariable model if the sample size for OCP use is 193, then the sample size for all other variables in the model is automatically diminished to 193. That is, 213 study participants (all males) are excluded from the analysis for all independent variables in the model. Therefore, the authors should re-do the analysis by excluding OCP use if the results of the present study are to be valid.

Response: We feel very sorry for not deleting the variable OCP use from “Table 3”. It was included by mistake. As per the previous comment we already removed OCP use from multivariable analysis and done the analysis again in the previous submission. 

9. The authors should still check the authors' guideline of PLoS ONE for the appropriate placement of tables within the manuscript.

Response: Comment accepted and changes have been made. Please see “Table 1”, “Table 2” and “Table 3”.

10. The discussion still needs to be more critical.

Response: Comment accepted and changes have been made.

11. The authors should make sure that all references are properly written.

Response: Comment accepted and modifications have been made. 

12. The data availability statement should be in line with PLoS ONE’s data availability policy.

Response: Comment accepted and changes have been made. Please see the highlighted manuscript line (358).

---

## [Decision Letter · Decision Letter 3]

13 Oct 2021

Heavy khat (Catha edulis) chewing and dyslipidemia as modifiable hypertensive risk factors among patients in Southwest, Ethiopia: Unmatched case-control study

PONE-D-20-40692R3

Dear Dr. Gebremedhin,

We’re pleased to inform you that your manuscript has been judged scientifically suitable for publication and will be formally accepted for publication once it meets all outstanding technical requirements.

Kind regards,

Samson Gebremedhin, PhD

Academic Editor

PLOS ONE

Additional Editor Comments (optional):

Reviewers' comments:

Reviewer's Responses to Questions

**Comments to the Author**

1. If the authors have adequately addressed your comments raised in a previous round of review and you feel that this manuscript is now acceptable for publication, you may indicate that here to bypass the “Comments to the Author” section, enter your conflict of interest statement in the “Confidential to Editor” section, and submit your "Accept" recommendation.

Reviewer #1: All comments have been addressed

2. Is the manuscript technically sound, and do the data support the conclusions?

Reviewer #1: Yes

3. Has the statistical analysis been performed appropriately and rigorously? 

Reviewer #1: Yes

4. Have the authors made all data underlying the findings in their manuscript fully available?

Reviewer #1: Yes

5. Is the manuscript presented in an intelligible fashion and written in standard English?

Reviewer #1: Yes

6. Review Comments to the Author

Reviewer #1: (No Response)

7. PLOS authors have the option to publish the peer review history of their article (what does this mean?). If published, this will include your full peer review and any attached files.

Reviewer #1: **Yes: **Ayalew Astatkie

---

## [Editor Report · Acceptance letter]

18 Oct 2021

PONE-D-20-40692R3 

Heavy khat (Catha edulis) chewing and dyslipidemia as modifiable hypertensive risk factors among patients in Southwest, Ethiopia: Unmatched case-control study 

Dear Dr. Gebremedhin:

I'm pleased to inform you that your manuscript has been deemed suitable for publication in PLOS ONE. Congratulations! Your manuscript is now with our production department. 

Kind regards, 

on behalf of

Dr. Samson Gebremedhin 

Academic Editor

PLOS ONE